# A Unified Framework for Synaesthesia Analysis

**Kun Sheng[1], Zhongqing Wang[1] *, Qingqing Zhao[2], Xiaotong Jiang[1] and Guodong Zhou[1]**

[1]Natural Language Processing Lab, Soochow University
[2]Institute of Linguistics, Chinese Academy of Social Sciences
{ksheng_22,devjiang}@outlook.com, zhaoqq@cass.org.cn
{wangzq,gdzhou}@suda.edu.cn

## Abstract

Synaesthesia refers to the description of perceptions in one sensory modality through concepts from other modalities. It involves not only a linguistic phenomenon, but also a cognitive phenomenon structuring human thought and action, which makes understanding it challenging. As a means of cognition, synaesthesia is rendered by more than sensory modalities, cue and stimulus can also play an important role in expressing and understanding it. In addition, understanding synaesthesia involves many cognitive efforts, such as identifying the semantic relationship between sensory words and modalities. Therefore, we propose a unified framework focusing on annotating all kinds of synaesthetic elements and fully exploring the relationship among them. In particular, we introduce a new annotation scheme, including sensory modalities as well as their cues and stimuli, which facilitate understanding synaesthetic information collectively. We further design a structure generation model to capture the relations among synaesthetic elements and generate them jointly. Through extensive experiments, the importance of the proposed dataset can be verified by the statistics and progressive performances. In addition, our proposed model yields state-of-the-art results, demonstrating its effectiveness.

## 1 Introduction

Synaesthesia, based on the Greek roots 'syn' (together) and 'aisthesia' (perception), describes a situation in which perceptions in different sensory modalities are associated in both perceptual experiences and verbal expressions (Cytowic, 1993; Popova, 2008; Shen and Aisenman, 2008). Synaesthesia in verbal expressions is a language usage whereby lexical items in one sensory modality are employed to describe perceptions in another sensory modality (Zhao et al., 2018; Zhao,

_______________
\* Corresponding author

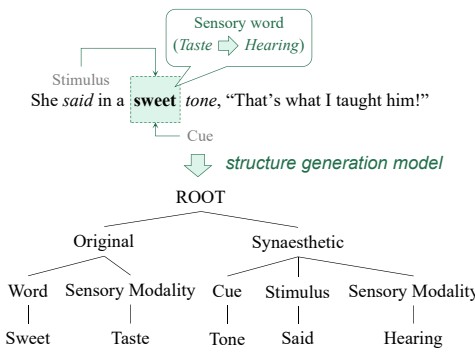

Figure 1: An example of the proposed unified framework for synaesthesia annotation and generation.

2020). For instance, as shown in Figure 1, the gustatory adjective "sweet" can be used to describe an auditory perception, as in the phrase "said in a sweet tone".

There are extensive studies on linguistic synaesthesia from various perspectives (Shen, 1997; Winter, 2019; Zhao et al., 2022). Nevertheless, synaesthesia has received little attention in natural language processing. As one of the initial study, Jiang et al. (2022) constructed a human-annotated Chinese synaesthesia dataset, where sensory words, original and synaesthetic sensory modalities are annotated. They further designed a pipeline system to identify the sensory word, and to detect the original and synaesthetic sensory modalities for the sensory word.

As a means of cognition and communication, synaesthesia is rendered by more than sensory modalities, cue and stimulus can also contribute to synaesthetic conceptualization. Cue and stimulus are the trigger words or expressions that cause the synaesthetic usages. As shown in Figure 1, since the cue "tone" and the stimulus "said" are both auditory words, it is easy to find that "sweet" is transferred from _taste_ to _hearing_. In addition, understanding synaesthesia involves many cognitive efforts, such as identifying the semantic re-

lationship between sensory words and modalities, as well as interpreting sensory modalities with the cue and the stimulus, which might be difficult for computers to deal with.

In this study, we propose a unified framework to address the above challenges by annotating all kinds of synaesthetic elements and fully exploring the relationship among them. In particular, we firstly extend the previous dataset by collecting more samples from social media. We then propose a new annotation framework, including sensory modalities as well as their cues and stimuli, which facilitate understanding synaesthesia information and help to improve the performance of automatic synaesthesia comprehension systems. Furthermore, we propose a structure generation model to capture the relations among synaesthetic elements, and generate all elements jointly. Afterwards, we employ a structure acquisition and composition based framework to improve the generation model by capturing the interdependency among synaesthetic elements.

The statistics show the potential and usefulness of the new annotation scheme. In addition, the experimental results demonstrate the effectiveness of the proposed model.We release our dataset named CSyna at https://github.com/dinoksaur/csyna.

## 2 Related Works

Studies on linguistic synaesthesia from a linguistic perspective focus on the directionality patterns and underlying mechanisms for synaesthetic transfers between different modalities. Note that "synaesthesia" and "synesthesia" are used interchangeably in the literature. For consistency, we use "synaesthesia" in this paper. For instance, previous studies (Ullmann, 1957; Williams, 1976; Strik Lievers, 2015; Zhao et al., 2019) found that the transfers of linguistic synaesthesia conformed to certain patterns, rather than mapping randomly. In terms of the mechanisms underlying synaesthetic transfers, Zhao et al. (2018) and Winter (2019) have suggested that linguistic synaesthesia is grounded in multiple mechanisms. In addition, Strik Lievers et al. (2013) and Strik Lievers and Huang (2016) focus on identifying linguistic synaesthetic expressions in natural languages. However, their studies are conducted by semi-automatic methods with lots of manual strategies. There are no comprehensive computational models with automatic synaesthesia detection em-ployed in previous methods.

Recently, Jiang et al. (2022) constructed a human-annotated Chinese synaesthesia dataset. They further proposed a pipeline system to identify sensory words, and to detect the original and synaesthetic sensory modalities. However, the annotated samples in their dataset is limited, and cue and stimulus are missing in their annotation scheme. Therefore, we propose a unified framework focusing on annotating all kinds of synaesthetic elements and fully exploring the relationship among them. In particular, we propose a new annotation scheme, including sensory modalities as well as their cues and stimuli, which are helpful for understanding synaesthesia information. We further employ a structure generation model to fully explore the relations between synaesthetic elements and to generate them jointly.

## 3 Data Annotation and Analysis

In this section, we first give a new scheme of synaesthesia. We then show the collection and annotation process of the dataset. After that, we give some fundamental statistics and analyses.

### 3.1 Scheme of Annotation

In this study, we propose a unified annotation scheme, which aims to investigate the capabilities of automatic systems not only to distinguish between sensory modalities, but also to capture their semantic constituents. Following the task of semantic role labeling (Gildea and Jurafsky, 2002), we aims to answer the question as "Who express What sensory modalities, towards Whom and Why?" (Màrquez et al., 2008; Campagnano et al., 2022). We thus take a subset of semantic roles, namely, cue and stimulus, to identify who or what the participants are in an action or event denoted by a sensory modality. The following is the description of the annotation scheme:

- **Sensory Word** is an adjective that expresses a sensory modality in a sentence.

- **Original Sensory Modality** is the original domain of a sensory word, which is always stationary, and not influenced by the context of the sensory word. The five 'Aristotelian' senses (Strik Lievers, 2015; Winter, 2019; Zhao, 2020), including Touch, Taste, Smell, Vision, and Hearing, are used in this study.

- **Synaesthetic Sensory Modality** is the actual sensory modality of a sensory word, which is influenced by the sensory word and its context.

- **Cue** is a trigger word or an expression modified by a sensory word.

- **Stimulus** is an entity, action or event that causes the synaesthetic sensory modality identified by the *cue*.

As shown in Figure 1, the sensory word "sweet" expresses the synaesthetic sensory modality *hearing*, which is different from its original sensory modality *taste*. In addition, the synaesthetic sensory modality is caused by its cue "tone" and stimulus "said". Therefore, the new scheme gives a comprehensive study of synaesthesia, which can be used to explore what and why synaesthesia would happen in a sentence. Additionally, it facilitates understanding synaesthesia information and helps to improve the performance of automatic synaesthesia comprehension systems.

## 3.2 Data Collection and Annotation

With the goal of creating a large-scale synaesthesia dataset to support research on understanding synaesthesia, we extend Jiang et al. (2022)'s dataset by collecting more samples from social media. Each sentence is ensured to contain at least one sensory adjective as the candidate sensory sentence. The detail statistic of the dataset can be found in Section 3.3.

Following the new scheme, the annotation of synaesthesia includes the original sensory modality, the synaesthetic sensory modality, the sensory word, the cue and the stimulus. In particular, we invited expert annotators to complete this challenging annotation task, which required a relatively deep understanding of synaesthetic units. The annotator team comprised five annotators who are postgraduate student researchers majoring in computational linguistics with synaesthesia or metaphor study backgrounds. The annotators formed groups of two, plus one extra person. We used cross-validation in annotation process: the two-member groups annotated, and the fifth person intervened if they disagreed.

Annotations of synaesthesia with multiple information rely on annotators' introspection, which might be subjective. Therefore, we added a guideline course, detailed instructions, and many sam-

| #Sentence | 24,000 |
|---|---|
| #Sensory Word | 236 |
| #Cue | 2,397 |
| #Stimulus | 706 |
| #Avg. Cue Per Sentence | 0.994 |
| #Avg. Stimulus Per Sentence | 0.304 |

Table 1: A summary statistics of the dataset.

| | Original | Synaesthetic |
|---|---|---|
| Vision | 4,901 | 5,103 |
| Hearing | 38 | 5,221 |
| Taste | 3,042 | 240 |
| Touch | 3,976 | 1,194 |
| Smell | 43 | 242 |

Table 2: Statistics of original and synaesthetic sensory modalities.

ples. We also held regular meetings to discuss annotation problems and matters that needed attention. The guidelines changed three times when new problems emerged or good improvement methods were found. The kappa score was used to measure inter-annotator agreements (Fleiss, 1971). The agreement on identification of sensory modality was $k = 0.78$; detection of cue and stimulus was $k = 0.71$, which means they are substantially reliable.

## 3.3 Statistic and Analysis

We firstly give the summary statistics of the dataset in Table 1. There are 24,000 sentences with 236 sensory words in the proposed new dataset, which is much larger than the previous dataset (Jiang et al., 2022). In addition, we also annotate 2,397 cues and 706 stimuli in the dataset. On average, each sentence includes 0.994 cue and 0.304 stimulus. It shows that most of sentences contain at least one cue or stimulus.

We then analyze the distribution of original and synaesthetic sensory modalities in Table 2. Among the five sensory modalities, vision and touch are the most frequent original sensory modalities. In addition, although synaesthesia rarely occurs in the auditory modality, the visual modality tends to transfer to the auditory modality. Thus, synaesthetic sentences with the auditory modality have the largest number (The examples can be found in Figure 2).

Furthermore, we give the transfer probability between the original and synaesthetic modalities,

| Original Sensory Modality | Synaesthetic Sensory Modality | Transfer Probability | Sensory Word | Cue | Stimulus | Example |
|---|---|---|---|---|---|---|
| Vision | Hearing | 74.13% (3633/4901) | big 大 | sound 聲 | shouted 喊叫 | Seeing with a face full of rage and anger, he shouted with big sound. 只見他滿臉怒容，氣極敗壞的大聲喊叫著. |
| | | | low 低 | shout 吼 | said 說 | she said to Michael in a low shout. 她近乎低吼地向麥可說. |
| | Touch | 21.14% (1036/4901) | tightly 緊 | eyes 眼睛 | closed 閉 | I closed my eyes tightly and bit my lower lip. 我緊閉著眼睛，死咬著下唇. |
| | | | loose 鬆 | floor 樓板 | cracked 裂 | The floor has long been loose and cracked. 樓板早已鬆裂. |
| Touch | Vision | 61.54% (2447/3976) | sharp 銳利 | gaze 目光 | sweeping 掃視 | I felt that he was sweeping the mountains with a sharp gaze. 我感到他正用銳利的目光掃視著山野. |
| | | | pointy 尖 | mouth 嘴 | saw 看到 | We saw a dog with big eyes and a pointy mouth. 我們看到了一隻大眼睛、尖嘴的狗. |
| | Hearing | 33.4% (1328/3976) | soft 輕 | Voice 一聲 | called out 叫 | Anna stood up and called out in a soft voice. 安娜站起身，一聲輕叫起來. |
| | | | harsh 凌厲 | voice 聲音 | heard 聽見 | I heard the irritated and harsh voice of the flight attendant. 我聽見空服員惱怒而凌厲的聲音. |
| Taste | Vision | 83.86% (2551/3042) | fresh 鮮 | yellow 黃 | appearance 外形 | Cologne phone booths are fresh yellow and have a very streamlined appearance. 科隆電話亭是鮮黃的，外形都很流線 |
| | | | mild 淡淡 | green 綠色 | looks 看 | From a distance, the river looks with a mild green color. 遠遠看去河水帶著淡淡的綠色. |

Figure 2: Transfer probabilities and examples between original and synaesthetic modalities.

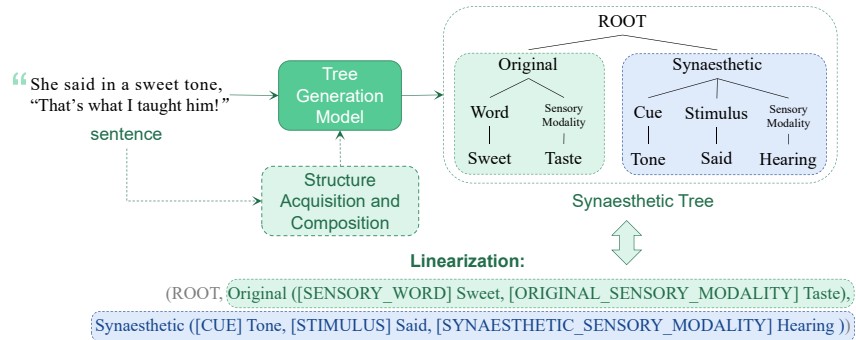

Figure 3: Overview of the synaesthetic structure generation model.

along with examples in Figure 2. Since there are too many combinations, we just show the most frequent original and synaesthetic pairs in the figure. Based on the synaesthesia transfer probability, we find that: tactile adjectives are the most likely to be used for vision, with the transfer probability of 61.5%. The association between vision and hearing is similar to that between touch and vision. Specifically speaking, visual adjectives are the most likely to be associated with hearing. Meanwhile, we also find that the synaesthetic sensory modalities are deeply influenced by the cue and stimulus. For example, although "harsh" is a tactile adjective, "voice" and "hear" trigger it to express hearing. In addition, "appearance" makes the sensory modality of "fresh" transferring from taste to vision.

## 4  Synaesthetic Structure Generation

Given a sentence, we employ a *structure generation model* to generate all the synaesthetic ele-

ments from a sentence. As shown in Figure 3, we firstly convert all the synaesthetic elements into a tree structure. We then employ a tree generation model to explore the relations among synaesthetic elements and generate them jointly. Afterwards, we utilize a structure acquisition and composition based framework to improve the structure generation model by capturing the interdependency among synaesthetic elements. In this section, we give the description of synaesthetic tree construction and the structure tree generation model, and then discuss the structure acquisition and composition framework in the next section.

### 4.1  Synaesthetic Tree Construction

As shown in the right side of Figure 3, the synaesthetic tree models a sentence using a rooted directed acyclic graph, highlighting its main elements (e.g. sensory word, cue, stimulus) and relations. Given a sentence, we convert synaesthetic elements into the synaesthetic tree as below:

- We create the original and synaesthetic node to represent the original and synaesthetic sensory information respectively, and connect them with a virtual root node;

- The sensory word and the original sensory modality are linked to the original node as leaves.

- The synaesthetic sensory modality, the cue and the stimulus are linked to the synaesthetic node as leaves.

Therefore, the tree structure is very important for understanding synaesthesia information and learning the correlations among synaesthetic elements. For instance, the connections between the sensory word and the original sensory modality can be used to identify the sensory word. In addition, the sub-tree of the synaesthetic sensory modality is helpful for detecting the synaesthetic sensory modality based on the cue and the stimulus.

Since it is much easier to generate a sequence than a tree (Vinyals et al., 2015; Lu et al., 2021), we linearize the synaesthetic tree to the target sequence. As shown in the bottom of Figure 3, we linearize a tree into a token sequence via depth-first traversal, where "(" and ")" are structure indicators used to represent the structure of linear expressions.

## 4.2 Synaesthetic Tree Generation

We then employ a text generation model to generate the linearized synaesthetic tree from the given sentence.

In this study, we employ a sequence-to-sequence model to generate the synaesthetic tree via a transformer-based encoder-decoder architecture (Vaswani et al., 2017). Given the token sequence $x = x_1, ..., x_{|x|}$ as input, the sequence-to-sequence model outputs the linearized representation $y = y_1, ..., y_{|y|}$. To this end, the sequence-to-sequence model first computes the hidden vector representation $H = h_1, ..., h_{|x|}$ of the input sentence via a multi-layer transformer encoder:

$$H = \text{Encoder}(x_1, ..., x_{|x|}) \quad (1)$$

where each layer of Encoder is a transformer block with the multi-head attention mechanism.

After the input token sequence is encoded, the decoder predicts the output tree structure token-by-token with the sequential input tokens' hidden

vectors. At the $i$-th step of generation, the self-attention decoder predicts the $i$-th token $y_i$ in the linearized form, and decoder state $h_i^d$ as:

$$y_i, h_i^d = \text{Decoder}([H; h_1^d, ..., h_{i-1}^d], y_{i-1}) \quad (2)$$

where each layer of Decoder is a transformer block that contains self-attention with decoder state $h_i^d$ and cross-attention with encoder state $H$.

The generated output structured sequence starts from the start token "$\langle bos \rangle$" and ends with the end token "$\langle eos \rangle$". The conditional probability of the whole output sequence $p(y|x)$ is progressively combined by the probability of each step $p(y_i|y_{<i}, x)$:

$$p(y|x) = \prod_{i=1}^{|y|} p(y_i|y_{<i}, x) \quad (3)$$

where $y_{<i} = y_1...y_{i-1}$, and $p(y_i|y_{<i}, x)$ are the probabilities over target vocabulary $V$ normalized by softmax.

Since all tokens in linearized representations are also natural language words, we adopt the pre-trained language model T5 (Raffel et al., 2020) as our transformer-based encoder-decoder architecture. In this way, the general text generation knowledge can be directly reused.

## 4.3 Objective Functions and Training

In this subsection, we show the objective functions and the training process of the proposed model.

The goal is to maximize the output linearized tree $X_T$ probability given the sentence $X$. Therefore, we optimize the negative log-likelihood loss function:

$$\mathcal{L} = -\frac{1}{|\tau|} \sum_{(X, X_T) \in \tau} \log p(X_T|X; \theta) \quad (4)$$

where $\theta$ is the model parameters, and $(X, X_T)$ is a (*sentence, tree*) pair in training set $\tau$, then

$$\log p(X_T|X; \theta) =$$
$$= \sum_{i=1}^{n} \log p(x_T^i|x_T^1, x_T^2, ...x_T^{i-1}, X; \theta) \quad (5)$$

where $p(x_T^i|x_T^1, x_T^2, ...x_T^{i-1}, X; \theta)$ is calculated by the decoder.

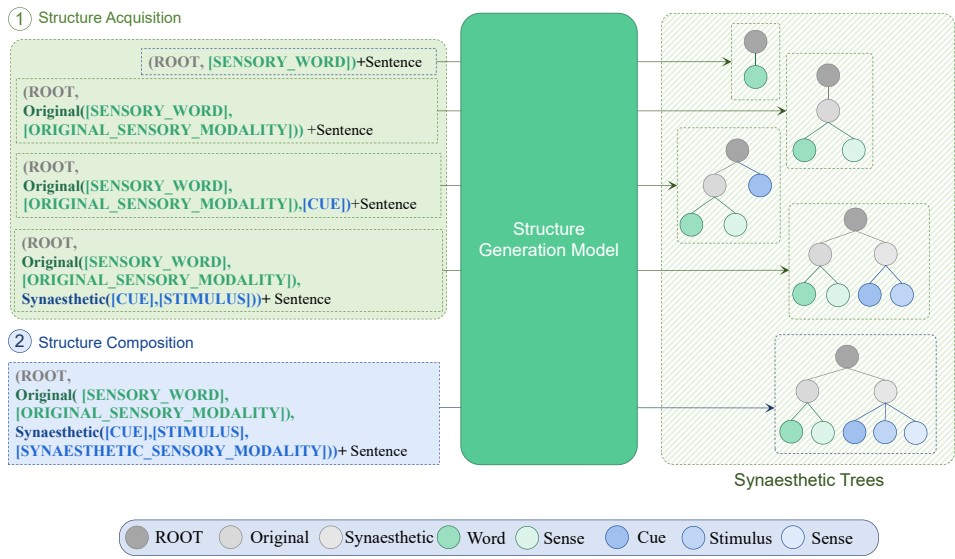

Figure 4: An example of structure acquisition and composition.

## 5 Structure Acquisition and Composition

Since the alignment between sentences and synaesthetic elements, and the interdependency among synaesthetic elements are very important for synaesthesia analysis, we then propose a novel strategy to improve the structure generation model by addressing the above issues. As shown in Figure 4, we firstly decompose the original task into several subtasks. Each subtask corresponds to mapping the natural language sentence to one or more synaesthetic elements. Afterwards, we feature a prompt-based learning strategy to separately acquire the structural knowledge of subtasks and employ the learned knowledge to tackle the main task, i.e., generating all the synaesthetic elements. In the structure acquisition stage, we train the model with all the subtasks in a multi-task learning manner; in the structure composition stage, we fine-tune the model with the main task to combine the acquired structural knowledge of subtasks and learn the dependency between them.

### 5.1 Structure Acquisition

As shown in Figure 4, we decompose the synaesthetic structure generation task into four subtasks. Basically, a subtask aims to translate the sentence to one or more synaesthetic elements. For example, the $SENSORY\_WORD$ subtask aims to generate the sensory word given the sentence.

We then train the sequence-to-sequence model with all subtasks using multi-task learning. We assign each subtask a special token, which is also used to denote its corresponding synaesthetic el-

ements. Then we construct a task prompt for each subtask based on the elements it contains. For example, The special token corresponding to $SENSORY\_WORD$ is "[ SENSORY_WORD ]". The input for each subtask simply adds a task prompt to the input for the original task.

### 5.2 Structure Composition

Training the model with multiple subtasks cannot capture the interdependency between them. In this stage, we fine-tune the model with the main task, i.e., generating all the synaesthetic elements to capture such information. As shown in Figure 4, we combine the prompts of subtasks to construct the prompt of the main task to guide the model to composite the knowledge of subtasks.

## 6 Experiments

In this section, we conduct various experiments to investigate the effectiveness of the proposed method on the synaesthetic elements generation task. In addition, we give several analyses and discussions to show the importance of the proposed new dataset.

### 6.1 Setting

We evaluate our proposed structure generation model on the new Chinese synaesthesia dataset. There are totally 24,000 sentences in the proposed new dataset. We split the dataset into the training set (80%), the validation set (10%) and the test set (10%). It should be noted that these sets are separated by sensory words, which means that the

| Method | Sensory Word | Original | Synaesthetic | Cue | Stimulus | Overall |
|--------|--------------|----------|--------------|-----|----------|---------|
| BERT+CRF | 74.6 | 74.2 | 70.3 | 77.4 | 83.9 | 53.0 |
| Radical | 75.6 | 75.1 | 71.4 | 77.7 | 84.2 | 54.3 |
| BART | 72.5 | 68.4 | 63.8 | 73.5 | 79.3 | 42.1 |
| T5 | 72.8 | 69.4 | 65.2 | 71.3 | 80.1 | 40.7 |
| Human | 77.1 | 75.0 | 73.6 | 79.2 | 80.4 | 49.1 |
| ChatGPT | 41.8 | 22.3 | 27.0 | 35.0 | 49.5 | 15.3 |
| Ours | 78.3 | 78.3 | 73.9 | 79.7 | 86.3 | 61.9 |

Table 3: Comparison with baselines.

sensory words among different sets are totally different.

We employ T5[1] and fine-tune its parameters for generation models. We tune the parameters of our models by grid searching on the validation dataset. We select the best models by early stopping using the Accuracy results on the validation dataset. The model parameters are optimized by Adam (Kingma and Ba, 2015) with a learning rate of 1e-4. The batch size is 16. Our experiments are carried out with an Nvidia RTX 3090 GPU. The experimental results are obtained by averaging ten runs with the random initialization.

We use F1-score as the evaluation metric for the sensory word, cue, and stimulus extraction, and weighted F1-score (Manning and Schütze, 1999) as the evaluation metric for the sensory modality detection. In addition, a synaesthetic quintet (sensory word, original sensory modality, synaesthetic sensory modality, cue, stimulus) is viewed as correct if and only if the five elements, as well as their combinations, are exactly the same as those in the gold quintet. On this basis, we calculate F1-score (Cai et al., 2021) as the *overall* evaluation metric for the synaesthesia analysis.

### 6.2 Main Results

We firstly compare the proposed structure generation model with various strong baselines on Table 3, where,

- **BERT+CRF** adopts BERT (Devlin et al., 2019) as the textual encoder, followed by a CRF output layer (Radford and Narasimhan, 2018).

- **Radical** (Jiang et al., 2022) is a pipeline system, which employs a radical-based neural network model to identify the sensory word's boundary and to jointly classify the original and synaesthetic sensory modalities.

- **BART** employs the pre-trained language model BART (Lewis et al., 2020) to generate the synaesthetic elements individually[2].

- **T5** utilizes T5 (Raffel et al., 2020) to generate the synaesthetic elements individually, which can be considered as the baseline method of the proposed model.

- **Human** engaged two volunteers to identify and label the synaesthetic elements, and the results are considered as representative of human performance.

- **ChatGPT** is a sibling model to Instruct-GPT (Ouyang et al., 2022), which is trained to follow an instruction in a prompt and provides a detailed response. We ask it to generate the synaesthetic elements from the input sentences[3].

From the results, we find that the basic generation model (i.e., BART, T5) cannot give an acceptable result, the performance of which is even lower than BERT+CRF. It may be due to that the basic generation model generates each element one by one, ignoring the correlations among them. In addition, the performance of ChatGPT is much lower than other models. One of the possible reasons is that we do not fine-tune it on the training data, and we only give some prompts to let it understand the synaesthesia analysis task. Based on analyzing its outputs, we find that it is hard for ChatGPT to predict the original and synaesthetic sensory modalities, and it also cannot capture the relations among all these synaesthetic elements.

---

[1]https://huggingface.co/t5-base

[2]https://huggingface.co/facebook/bart-base

[3]https://openai.com/chatgpt

| Method | F1-score |
|--------|----------|
| Ours | 61.9 |
| -SAC | 60.7 |
| -Tree | 60.5 |
| -SAC -Tree | 59.8 |

Table 4: Impacts of different factors in the proposed structure generation model with overall F1-score measurements.

| Tuples | $w$ | $o$ | $s$ | $c$ | $t$ |
|--------|-----|-----|-----|-----|-----|
| $(w)$ | 71.3 | - | - | - | - |
| $(w, o)$ | 76.8 | 75.7 | - | - | - |
| $(w, s)$ | 76.9 | - | 71.2 | - | - |
| $(w, o, s)$ | 77.2 | 76.1 | 71.7 | - | - |
| $(w, s, c)$ | 77.5 | - | 72.7 | 77.8 | - |
| $(w, s, t)$ | 76.9 | - | 72.1 | - | 83.8 |
| $(w, s, c, t)$ | 78.3 | - | 73.1 | 77.7 | 84.8 |
| Ours | 78.3 | 78.3 | 73.9 | 79.7 | 86.3 |

Table 5: Effects of synaesthetic elements.

In contrast, our proposed model outperforms all the previous studies significantly ($p < 0.05$) in all settings. Compared to native speakers, our proposed model surpasses human performance, making it easier to generate all five synaesthetic elements correctly. This indicates that the tree structure is much more helpful for capturing the correlations among synaesthetic elements. Furthermore, the results also indicate the effectiveness of the structure acquisition and composition strategy, which is used to learn alignment between sentences and synaesthetic elements, and the interdependency among synaesthetic elements. The details of the case study are shown in Appendix A.

### 6.3 Impact of Different Factors

As shown in Table 4, we employ ablation experiments to analyze the impact of different factors in the proposed structure generation model.

If we remove the structure acquisition and composition strategy (-SAC), the performance drops to 60.7%. It indicates that this strategy is very important for generating a valid tree and capturing the interdependency among synaesthetic elements. If we remove the tree structure, and only generate the flat sequence, the performance drops to 60.5%. It shows that the tree structure is beneficial to capture the correlations between synaesthetic elements. Furthermore, we also find that the proposed model is much more effective than the simple text generation model. If we remove both the structure acquisition and composition strategy and the tree structure, the proposed model degrades to a T5-based sequence-to-sequence model, and the performance drops to 59.8%.

### 6.4 Effects of Synaesthetic Elements

We then analyze the effect of synaesthetic elements in Table 5, where $w$ denotes the sensory word, $o$ and $s$ are the original and synaesthetic sensory modalities respectively, and $c$ and $t$ are

the cue and the stimulus respectively. In addition, the rows in the table mean different combinations of synaesthetic elements. For example, $(w, s)$ means that we used the proposed structure generation model to generate the sensory word and the synaesthetic sensory modality jointly.

From the table, we can find that: 1) The sensory modalities are deeply correlated with the sensory words. If we jointly generate the sensory word with original $(w, o)$ or synaesthetic sensory modalities $(w, s)$, the performance of the sensory word extraction is improved more than 5%. 2) The original and synaesthetic sensory modalities are correlated, and the joint prediction $(w, o, s)$ is better than predicting them individually. 3) Cue $c$ and stimulus $t$ are both helpful for extracting the sensory word and predicting the synaesthetic sensory modality. If we integrate them into the generation model, the performance of the sensory word extraction and the synaesthetic sensory modality prediction are higher than before. 4) When we generating all the elements jointly (ours), the performance of the proposed model can achieve the best performance.

In summary, the above observations show that all the synaesthetic elements are correlated, and the structure generation model is helpful for generating them jointly. Furthermore, the results among subtasks (i.e., $(w, o)$, $(w, s, c)$) and the main task (ours) also indicate the importance of the structure acquisition and composition strategy, which is used to learn from all the subtasks to the main task.

## 7 Conclusion

Synaesthesia refers to the description of perceptions in one sensory modality through concepts from other modalities. It involves not only a linguistic phenomenon, but also a cognitive phenomenon structuring human thought and action, which makes understanding it challenging and re-

warding. In this study, we give a comprehensive study on the synaesthesia analysis from a computational perspective. In particular, we firstly introduce a new annotation framework, including annotating the sensory modality as well as their cues and stimuli, which facilitates understanding the synaesthesia information. We further design a structure generation model to fully explore the relations among synaesthetic elements and to generate them jointly. The statistics show the potential and usefulness of the proposed new dataset. In addition, the experimental results demonstrate the effectiveness of the proposed model.

## Limitation

Our work focuses on employing a unified framework for the synaesthesia analysis. However, some more general topics should be addressed as future work, such as the metaphor and sensory modality analysis. Another limitation is the computational complexity of the proposed model, whose training time is slower than the classification-based models.

## Acknowledgements

We would like to thank the anonymous reviewers for their insightful and valuable comments. Also, we gratefully acknowledge funding from the National Natural Science Foundation of China (No. 62076175, No.61976146), and Jiangsu Innovation Doctor Plan.

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

## A Case Study

| | w | o | s | c | t |
|---|---|---|---|---|---|
| **Example1:** Together, every person responded with warm sound of hitting hands. 大家一同報以熱烈的掌聲. | | | | | |
| **Baseline** | every 大 ✘ | Vision ✘ | Hearing | sound 聲 ✘ | - |
| **Ours** | warm 熱烈 ✔ | Touch ✔ | Hearing | sound of hitting hands 掌聲 ✔ | - |
| **Example2:** Huahui suddenly became angry, his face turned red, and he exclaimed with big sound. 華輝突然發怒，脹紅了臉，大聲道. | | | | | |
| **Baseline** | turned 脹 ✘ | Touch ✘ | Hearing | sound 聲 | exclaimed 道 |
| **Ours** | big 大 ✔ | Vision ✔ | Hearing | sound 聲 | exclaimed 道 |
| **Example3:** Longfei said with a bitter expression on the face. 龍飛苦著臉道. | | | | | |
| **Baseline** | bitter 苦 | Taste | Hearing ✘ | face 臉 | said 道 ✘ |
| **Ours** | bitter 苦 | Taste | Vision ✔ | face 臉 | - ✔ |

Figure 5: The examples of case study. True predictions are marked with ✔ while false predictions are marked with ✘.

We provide case studies in Figure 5, wherein we select three examples to demonstrate the impact of the proposed model compared with baselines.

The first example and the second example are both about the effect of capturing the interdependency between sensory words and cues. With the help of tree structure and structure acquisition and composition, the proposed model can generate accurate sensory words and cues. Sensory word modifies cue, and the results of baseline models do not adhere to this rule.

The third example pertains to the connection between stimulus and synaesthetic sensory modality. Although the other three synaesthetic ele-

ments are accurate, baseline models fail to comprehend that the stimulus "said" does not refer to the synaesthetic sensory modality indicated by the cue "face".