# OpenReview forum: "A Unified Framework for Synaesthesia Analysis"
_EMNLP/2023/Conference — EMNLP 2023 Findings_

### Official Review · Reviewer_oMzF · 2023-07-26

**Soundness:** 2

**Excitement:**

3: Ambivalent: It has merits (e.g., it reports state-of-the-art results, the idea is nice), but there are key weaknesses (e.g., it describes incremental work), and it can significantly benefit from another round of revision. However, I won't object to accepting it if my co-reviewers champion it.

**Paper Topic And Main Contributions:**

This work aims to generate synaesthesia annotations for sentences. First, it builds a synaesthesia corpus. Then, it converts the synaesthesia data into a tree structure and linearizes the tree. Finally, the sentence-to-synaesthesia tree generation is learned by the T5 pre-trained model. In particular, the author proposes to transform the task of generating whole synaesthesia data into multiple subtasks to learn different parts of data to improve accuracy.

**Questions For The Authors:**

1. Did previous synesthesia detection works only use bert-like classification models? Are there any works that employ generative models?

2. How can synesthesia, cue, and other outputs be placed in the baseline models? Why is there such a performance difference between the proposed method based on T5 and baselines BART, T5?


**Reasons To Accept:**

1. The paper is well-written and easy to understand.

2. For the synesthesia detection task, it is a reliable idea to use the generative model T5 to generate Synesthesia, cue, stimulus, etc.

3. Compared with previous work, this work increases the detection of cues and stimulus.

**Reasons To Reject:**

1. The method is solid, but generating synesthesia for sentences via T5 seems less innovative. Although I read the related work [1] mentioned in this article, only using bert for synesthesia classification, not a generative model.

2. The main contribution of this work may be the constructed corpus. It is noted that the work does not mention that data will be open-sourced, and the related work mentioned in this work does not open source data too, which leads to the incompatibility of datasets and the inability to compare methods. This seems to be detrimental to research in the community.

3. The author doesn't mention how to place synesthesia, cue and other outputs in the baseline. Note that baselines include generative models such as BART and T5, and the proposed method also uses T5. Since all methods require flattening the output data, why is there such a performance difference between the proposed method based on T5 and baselines BART, T5?

[1] Jiang et al., Chinese Synesthesia Detection: New Dataset and Models

**Reproducibility:**

2: Would be hard pressed to reproduce the results. The contribution depends on data that are simply not available outside the author's institution or consortium; not enough details are provided.

**Reviewer Confidence:**

3: Pretty sure, but there's a chance I missed something. Although I have a good feel for this area in general, I did not carefully check the paper's details, e.g., the math, experimental design, or novelty.

---

> ### Author Rebuttal · Authors · 2023-08-27
>
> Dear reviewer,
>
> We really appreciate your valuable reviews, we will respond to your review one by one as follows:
>
> > The method is solid, but generating synesthesia for sentences via T5 seems less innovative. Although I read the related work [1] mentioned in this article, only using bert for synesthesia classification, not a generative model.
>
> **Response:** Our main contribution in the model architecture is the idea of using a unified generative framework, which significantly surpasses all existed synaesthesia related work (Jiang et al., 2020). We only use T5 as a backbone, which can be easily replaced with other trending generative methods directly without big changes.
>
> > The main contribution of this work may be the constructed corpus. It is noted that the work does not mention that data will be open-sourced, and the related work mentioned in this work does not open source data too, which leads to the incompatibility of datasets and the inability to compare methods. This seems to be detrimental to research in the community.
>
> **Response:** For fair comparison, we use the same constructed corpus on all experiments. And this constructed dataset will be open-sourced. By extending the size of previous dataset and adding the annotation of cue and stimulus, the dataset we have proposed has been refined and will be open-sourced for collaborative research on synaesthetic phenomena.
>
> > The author doesn't mention how to place synesthesia, cue and other outputs in the baseline. Note that baselines include generative models such as BART and T5, and the proposed method also uses T5. Since all methods require flattening the output data, why is there such a performance difference between the proposed method based on T5 and baselines BART, T5?
>
> **Response:** Different from the flat sequence generated in Section 6.3 ablation experiment, the baseline BART and T5 generate each synaesthesia element one by one. The five individual synaesthetic element generation tasks do not capture the correlations among them, which is why the performance of basic generation model is even lower than that of BERT+CRF. In order to explore the advantages of co-generating synaesthesia elements, we choose the model of  generating synaesthetic elements individually as basic generation model.
>
> > Did previous synesthesia detection works only use bert-like classification models? Are there any works that employ generative models?
>
> **Response:** Yes, previous synesthesia detection works only use bert-like classification models. We are the first to employ generative models for synaesthesia analysis, so that we can better capture the relationship between synesthesia elements and generate them jointly.

---

### Official Review · Reviewer_q3wD · 2023-08-07

**Soundness:** 3

**Excitement:**

4: Strong: This paper deepens the understanding of some phenomenon or lowers the barriers to an existing research direction.

**Paper Topic And Main Contributions:**

The paper addresses the problem of Synaesthesia which refers to the description of perceptions in one sensory modality through concepts from other modalities. The paper proposes a unified framework for annotation of synaesthetic elements. Specifically, the paper introduces a new annotation scheme that aids in understanding synaesthetic information. In addition, the paper proposes a new model that firstly converts all the synaesthetic elements into a tree structure and then employs a tree generation model to explore the relations among synaesthetic elements and generate them jointly. Experimental results and analysis demonstrate the utility of the proposed benchmark as well as the model.

**Reasons To Accept:**

The paper is well-motivated and well-organized in terms of style and structure. The proposed annotation framework in section 3.1 is interesting and will be useful in fostering more research in comprehending synaesthesia information. The paper performs a detailed analysis in section 6.3 and 6.4 on the impact and effect of synaesthetic elements. Overall, the work is well-written and solid in terms of novelty, new benchmark and experiment analysis.

**Reasons To Reject:**

I do not find any major weaknesses with this work. Readability of section 4.3 could be improved with more intuitive explanations. Are the authors planning to make this dataset publicly available? Please clarify. I suggest adding more examples from the dataset (to appendix) to better understand the quality of the annotations. I am happy to revise my scores based on the author 's response in the rebuttal on the dataset release.

**Reproducibility:**

2: Would be hard pressed to reproduce the results. The contribution depends on data that are simply not available outside the author's institution or consortium; not enough details are provided.

**Reviewer Confidence:**

3: Pretty sure, but there's a chance I missed something. Although I have a good feel for this area in general, I did not carefully check the paper's details, e.g., the math, experimental design, or novelty.

---

> ### Author Rebuttal · Authors · 2023-08-27
>
> Dear reviewer,
>
> We really appreciate your valuable reviews, we will respond to your review one by one as follows:
>
> > Are the authors planning to make this dataset publicly available? Please clarify.
>
> **Response:** Yes, we will release the dataset. By extending the size of previous dataset and adding the annotation of cue and stimulus, the dataset we have proposed has been refined and can now be open-sourced for collaborative research on synaesthetic phenomena.
>
>
> > I suggest adding more examples from the dataset (to appendix) to better understand the quality of the annotations. I am happy to revise my scores based on the author 's response in the rebuttal on the dataset release.
>
> **Response:**  Thanks for your valuable suggestion, we will add more examples from the dataset to appendix in the camera-ready version.

---

### Official Review · Reviewer_UVPL · 2023-08-10

**Typos Grammar Style And Presentation Improvements:** NA
**Soundness:** 4

**Excitement:**

4: Strong: This paper deepens the understanding of some phenomenon or lowers the barriers to an existing research direction.

**Missing References:**

NA

**Paper Topic And Main Contributions:**

The authors presents an annotation framework and model for understanding and generating grapheme-color synaesthetic experiences. The framework identifies synaesthetic connections and incorporates diverse sensory cues, enriching comprehension. The model effectively generates synaesthetic elements by considering interdependencies between different components. Through social media data, the model demonstrates superior performance compared to baseline methods. This work contributes significantly to synaesthesia research by offering a nuanced annotation approach and a novel generation model for complex sensory experiences.

**Questions For The Authors:**

- Can you elaborate on the steps taken to ensure consistency and minimize subjectivity in annotating multiple synaesthetic elements, given the complexity of the annotation framework?
- The structure generation model appears to offer improved accuracy, but could you provide more insight into how the model's decisions can be interpreted or visualized to gain a better understanding of its inner workings?
- In the context of cognitive science and psychology, how might the insights gained from your study contribute to our understanding of the cognitive processes underlying synaesthesia and cross-modal perception?

Please also see the weaknesses.

**Reasons To Accept:**

- They introduces a new annotation framework that goes beyond previous studies by including sensory modalities, cues, and stimuli. This comprehensive approach provides a more holistic understanding of synaesthesia.
- The proposed structure generation model captures the relationships between synaesthetic elements and generates them jointly. This approach considers the interdependencies that exist between various elements, enhancing the accuracy of synaesthesia comprehension.
- The authors extends an existing dataset by collecting more samples from social media, resulting in a larger and more diverse dataset for training and evaluation.
- The paper presents detailed experimental results that clearly demonstrate the superiority of the proposed model over various baseline methods. The evaluation metrics provide a comprehensive view of model performance.

**Reasons To Reject:**

- While the comprehensive annotation framework is a strength, it also introduces complexity and potential subjectivity in annotating multiple synaesthetic elements. How were inter-annotator disagreements resolved, and what measures were taken to ensure consistency?
- The study focuses on Chinese synaesthesia, and the proposed methods might need adaptation to work effectively with other languages. How transferable is the proposed framework to synaesthesia in languages with different syntactic and semantic structures?
- The authors acknowledges that understanding synaesthesia involves cognitive efforts, which could be challenging for computers. How does the proposed model's performance compare to human performance in terms of identifying and generating synaesthetic elements?

**Reproducibility:**

3: Could reproduce the results with some difficulty. The settings of parameters are underspecified or subjectively determined; the training/evaluation data are not widely available.

**Reviewer Confidence:**

4: Quite sure. I tried to check the important points carefully. It's unlikely, though conceivable, that I missed something that should affect my ratings.

---

> ### Author Rebuttal · Authors · 2023-08-27
>
> Dear reviewer,
>
> We really appreciate your valuable reviews, we will respond to your review one by one as follows:
>
> > How were inter-annotator disagreements resolved, and what measures were taken to ensure consistency?
>
> **Response:** To resolve the inter-annotator disagreements and ensure consistency, we took the below annotation steps:
> 1) We first asked a linguistic expert to annotate a set of sentences as exemples.
> 2) We then used these examples as a guide to train the annotators.
> 3) In the process of trial annotation, we adjusted the guidelines in time when new problems emerged with the help of the linguistic expert.
> 4) To resolve the inter-annotator disagreement, we use cross-validation methods for annotation correction.
> 5) We use kappa score to measure agreements between annotators.
> 6) Additionally, regular meetings were held to further ensure annotation consistency.
>
> > How transferable is the proposed framework to synaesthesia in languages with different syntactic and semantic structures?
>
> **Response:**  In fact, our proposed framework can be transferred in other languages directly, since the synaesthesia phenomenon has common cross-lingual features in expressions [1] [2]. That is, synaesthesia phenomenon in nearly all languages is likely to have the roles in our dataset: stimulus, cue, sensory word and sensory modality.
>
> For instance, if we have an English synesthesia dataset, we can utilize it to train a pretrained English generative model within our proposed framework. This model will leverage its internal understanding of English syntax and semantics.
>
> [1] Zhao, Q. 2020. Embodied Conceptualization or Neural Realization: A Corpus-Driven Study of Mandarin Synaesthetic Adjectives. Springer, Singapore.
>
> [2] Strik Lievers, F. 2015. Synaesthesia: A corpus-based study of cross-modal directionality. Functions of language, 22(1):69–95.
>
> > How does the proposed model's performance compare to human performance in terms of identifying and generating synaesthetic elements?
>
> **Response:**   First, it should be noted that the task is challenging for not only computers but also human without expert knowledge since there are some difficult cases in synaesthesia phenomenon. And that’s why we want to use a computational linguistic method to do the task. That’s also the reason why our annotators need the guidance from a linguistic expert and regular meetings, even they are all Chinese native speakers. So, from the annotators’ experiences, we believe our methods can reach a comparable performance compared to a native speaker without any expert knowledge training.
>
> Second, it’s still a very good advice to compare the performances from human and the proposed model, we will add the human performance results in the camera-ready version.
>
> > The structure generation model appears to offer improved accuracy, but could you provide more insight into how the model's decisions can be interpreted or visualized to gain a better understanding of its inner workings?
>
> **Response:**  As a means of cognition and communication, synaesthesia is rendered by more than sensory modalities, cue and stimulus can also contribute to synaesthetic conceptualization. Previous methods fail to take all these roles into consideration, we use a tree structure to model the relationship among all roles in a more structural way. Moreover, we use a generative framework to capture the relations among synaesthetic
> elements, and generate all elements jointly.
>
> Specifically, since the alignment between sentences and synaesthetic elements, and the interdependency among synaesthetic elements are very important for synaesthesia analysis, we first design a structure acquisition stage to learn the alignment, then we design a structure composition stage to model the interdependency among synaesthetic elements.
>
> > In the context of cognitive science and psychology, how might the insights gained from your study contribute to our understanding of the cognitive processes underlying synaesthesia and cross-modal perception?
>
> **Response:**  The results of this study provide new evidence for lexical conceptual mappings of linguistic synesthesia proposed by Zhao et al. (2022). That is , linguistic synesthesia is also a type of conceptual metaphor, where lexicalized concepts of sensory properties are involved.
>
> Zhao, Q., Ahrens, K., and Huang, C.-R. 2022. Linguistic synesthesia is metaphorical: a lexical-conceptual account. Cognitive Linguistics, 33(3):553–583.

---

### Meta-Review · Area_Chair_UuLX · 2023-09-10

**Recommendation:** 3

**Metareview:**

The paper under consideration presents an innovative approach to understanding and generating grapheme-color synesthetic experiences. It introduces a comprehensive annotation framework and a novel model for this purpose. Overall, the work is well-structured, motivated, and addresses an important topic in synesthesia research. It makes significant contributions to the field, but there are also some potential concerns that need to be addressed.

Pros:

- The paper introduces a novel annotation framework that includes sensory modalities, cues, and stimuli, providing a holistic understanding of synesthesia.
- The proposed model effectively captures the interdependencies between synesthetic elements, enhancing comprehension accuracy.
- The expansion of the dataset through social media data collection contributes to a more diverse and robust dataset.
- The experimental results clearly demonstrate the superiority of the proposed model over baseline methods, providing a comprehensive view of its performance.

Cons:

- The comprehensive annotation framework may introduce complexity and subjectivity. The paper lacks information on how inter-annotator disagreements were resolved and how consistency was ensured.
- The study focuses on Chinese synesthesia, and the generalizability of the proposed framework to other languages with different structures is unclear. Moreover, the title does not specify that the study focusses on Chinese only
- The paper acknowledges that understanding synesthesia involves cognitive efforts. A comparison of the proposed model's performance to human performance in identifying and generating synesthetic elements is missing.
- Section 4.3 could be improved for better readability, and the authors should clarify their plans for making the dataset publicly available to foster research.
- While the paper is well-structured, the use of T5 for synesthesia generation might be considered less innovative in comparison to other generative models.

In summary, the paper makes notable contributions to synesthesia research with its comprehensive framework and model, but it should address concerns related to subjectivity in annotation, cross-lingual adaptability, and comparisons to human performance. Additionally, improvements in readability and dataset availability are recommended.

In the rebuttals, the authors were able to provide access to the data and explain many of the issues raised by the reviewers, yet the paper would need some substantial edits.

---

### Decision · Program_Chairs · 2023-10-07

**Decision:**

Accept-Findings

**Comment:**

The paper under consideration presents an innovative approach to understanding and generating grapheme-color synesthetic experiences. It introduces a comprehensive annotation framework and a novel model for this purpose. Overall, the work is well-structured, motivated, and addresses an important topic in synesthesia research. It makes significant contributions to the field, but there are also some potential concerns that need to be addressed.

Pros:

- The paper introduces a novel annotation framework that includes sensory modalities, cues, and stimuli, providing a holistic understanding of synesthesia.
- The proposed model effectively captures the interdependencies between synesthetic elements, enhancing comprehension accuracy.
- The expansion of the dataset through social media data collection contributes to a more diverse and robust dataset.
- The experimental results clearly demonstrate the superiority of the proposed model over baseline methods, providing a comprehensive view of its performance.

Cons:

- The comprehensive annotation framework may introduce complexity and subjectivity. The paper lacks information on how inter-annotator disagreements were resolved and how consistency was ensured.
- The study focuses on Chinese synesthesia, and the generalizability of the proposed framework to other languages with different structures is unclear. Moreover, the title does not specify that the study focusses on Chinese only
- The paper acknowledges that understanding synesthesia involves cognitive efforts. A comparison of the proposed model's performance to human performance in identifying and generating synesthetic elements is missing.
- Section 4.3 could be improved for better readability, and the authors should clarify their plans for making the dataset publicly available to foster research.
- While the paper is well-structured, the use of T5 for synesthesia generation might be considered less innovative in comparison to other generative models.

In summary, the paper makes notable contributions to synesthesia research with its comprehensive framework and model, but it should address concerns related to subjectivity in annotation, cross-lingual adaptability, and comparisons to human performance. Additionally, improvements in readability and dataset availability are recommended.

In the rebuttals, the authors were able to provide access to the data and explain many of the issues raised by the reviewers, yet the paper would need some substantial edits.